# A Disintegrin and Metalloproteinase 9 (ADAM9) in Advanced Hepatocellular Carcinoma and Their Role as a Biomarker During Hepatocellular Carcinoma Immunotherapy

**DOI:** 10.3390/cancers12030745

**Published:** 2020-03-21

**Authors:** Sooyeon Oh, YoungJoon Park, Hyun-Jung Lee, Jooho Lee, Soo-Hyeon Lee, Young-Seok Baek, Su-Kyung Chun, Seung-Min Lee, Mina Kim, Young-Eun Chon, Yeonjung Ha, Yuri Cho, Gi Jin Kim, Seong-Gyu Hwang, KyuBum Kwack

**Affiliations:** 1Chaum Life Center, CHA University School of Medicine, Seoul 06062, Korea; 2Department of Biomedical Science, College of Life Science, CHA University, Seongnam 13488, Korea; 3Center for Research & Development, CHA Advanced Research Institute, Seongnam 13488, Korea; 4Department of Gastroenterology, CHA Bundang Medical Center, CHA University School of Medicine, Seongnam 13496, Korea; 5Immunotherapy Development Team, R & D Division, CHA Biolab, Seongnam 13488, Korea; 6Department of Food Science and Biotechnology, College of Life Science, CHA University, Seongnam 13488, Korea; 7Department of Gastroenterology, CHA Gangnam Medical Center, CHA University School of Medicine, Seoul 06135, Korea

**Keywords:** hepatocellular carcinoma, a disintegrin and metalloprotease 9, nivolumab, natural killer, immunotherapy

## Abstract

The chemotherapeutics sorafenib and regorafenib inhibit shedding of MHC class I-related chain A (MICA) from hepatocellular carcinoma (HCC) cells by suppressing a disintegrin and metalloprotease 9 (ADAM9). MICA is a ligand for natural killer (NK) group 2 member D (NKG2D) and is expressed on tumor cells to elicit attack by NK cells. This study measured *ADAM9* mRNA levels in blood samples of advanced HCC patients (*n* = 10). In newly diagnosed patients (*n* = 5), the plasma *ADAM9* mRNA level was significantly higher than that in healthy controls (3.001 versus 1.00, *p* < 0.05). Among four patients treated with nivolumab therapy, two patients with clinical response to nivolumab showed significant decreases in fold changes of serum *ADAM9* mRNA level from 573.98 to 262.58 and from 323.88 to 85.52 (*p* < 0.05); however, two patients with no response to nivolumab did not. Using the Cancer Genome Atlas database, we found that higher expression of *ADAM9* in tumor tissues was associated with poorer survival of HCC patients (log-rank *p* = 0.00039), while *ADAM10* and *ADAM17* exhibited no such association. In addition, *ADAM9* expression showed a positive correlation with the expression of inhibitory checkpoint molecules. This study, though small in sample size, clearly suggested that *ADAM9* mRNA might serve as biomarker predicting clinical response and that the ADAM9-MICA-NKG2D system can be a good therapeutic target for HCC immunotherapy. Future studies are warranted to validate these findings.

## 1. Introduction

Hepatocellular carcinoma (HCC) is the fifth most prevalent cancer and the third leading cause of cancer associated mortality worldwide [1,2]. Advanced stage HCC accompanied with portal vein invasion, distant metastasis, or lymph node metastasis is hard to treat due to the underlying liver cirrhosis, frequent recurrence, or multiple occurrence [3,4]. Currently, the tyrosine kinase inhibitors (TKI) sorafenib and lenvatinib are the first-line treatments in advanced HCC [5,6,7,8]. Regorafenib, nivolumab, cabozantinib, and ramucirumab were approved as second line therapies following their recent successes in several clinical trials [9,10,11,12,13,14,15]. Despite these breakthroughs, many patients still do not survive advanced HCC.

Among the recent breakthroughs in HCC treatments, immunotherapy is particularly notable. As a leading therapeutic in immunotherapy targeting programmed cell death 1 protein (PD-1), nivolumab demonstrated a 20% objective response rate and a 64% disease control rate in HCC patients who progressed on sorafenib [16]. Adoptive cellular immunotherapy using natural killer (NK) cells, cytokine induced killer (CIK) cells, or dendritic cells have also been studied [3]. Some adoptive cellular immunotherapy treatment modalities exhibited survival benefit in HCC patients [17]. These encouraging results were attributed to the immunological characteristics of the liver, which shelters a large pool of immune cells belonging to both the innate and acquired immune systems. The interactions between HCC and immune cells play a major role in tumor escape from immune surveillance resulting in HCC progression [18]. Inadequate co-stimulation, failure of tumor associated antigens (TAA) processing and presentation by dendritic cells, along with the suppression of effector T and NK cells are proposed mechanisms by which the HCC tumor cells evade the host immune system [19]. At the same time, HCC is indicated as a hot tumor that is characterized by increased expression of checkpoint molecules such as PD-1 and PD-1 ligand 1 (PD-L1) proteins, a large pool of tumor infiltrating immune cells, and high tumor mutational burdens, giving rise to ample amount of TAAs [20]. These immunological features enable HCC to benefit from immunotherapy. Thus, harnessing these immune characters through combination immunotherapy is proposed as the next important step in treatment of advanced HCC [4,13,14,15,21,22]. 

In this regard, a disintegrin and metalloproteases 9 (*ADAM9*) pathway in relation to MHC class I-related chain A (MICA) may provide a strategic ground for combination immunotherapy. Cell membrane-bound MICA (mMICA) is a ligand for NK group 2 member D (NKG2D), a stimulatory receptor on NK cells. The mMICA expressed in human HCC cells signals NK cells and other immune cells to kill the transformed hepatoma cells [23,24]. However, ADAM9, a matrix metalloproteinase (MMP) expressed by cancer cells, cleaves mMICA, releasing soluble MICA (sMICA). This sMICA acts as a decoy, weakening the cytotoxic immunity provided by NK cells and CD8+ T cells. This mMICA shedding by *ADAM9* protease is identified as a mechanism of HCC escape from the host immune surveillance. Fortunately, sorafenib and regorafenib inhibit the expression of *ADAM9* mRNA [23,24], restoring the host immunity against HCC and generating a room for synergistic action by adoptive cell therapy with NK cells or CD8+ T cells. Thus, the ADAM9-MICA-NKG2D system may provide a strategic target for a novel chemoimmunotherapy combining adoptive NK cell therapy and sorafenib or regorafenib [24].

In this pilot observational study, we aimed to characterize *ADAM9* mRNA expression in blood samples of advanced HCC patients according to their clinical courses. To support our findings, we probed the role of *ADAM9* as a prognostic biomarker for HCC using the Cancer Genome Atlas (TCGA) database. Furthermore, we present the case of a patient who achieved complete remission with regorafenib and autologous NK cell combination immunotherapy.

## 2. Results

### 2.1. Patient Characteristics

This study was conducted in CHA Bundang Hospital between January 2017 and November 2019. Advanced HCC patients eligible for this study, who were to be treated with sorafenib, regorafenib, or nivolumab as standard-of-care therapy and met our inclusion and exclusion criteria, were invited to the present study. A total of 10 patients participated in this study. The demographic and clinical details of each participant are listed in Table 1.

The 10 participants comprised eight chronic hepatitis B (CHB), one chronic hepatitis C (CHC) and one non-viral HCC patients. At the time of enrollment, five participants (Subject No. (#) 1 to #5) were newly diagnosed and treatment-naïve; they were subjected to sorafenib as first-line therapy. In addition, four patients (Subject #7 to #10) had failed first-line sorafenib therapy and were subjected to nivolumab therapy, whereas one patient (Subject #6) was enrolled after the patient had already reached near complete remission.

The baseline characteristics of the participants and their HCC are summarized in Appendix A. The mean age was 57.4 years, and 80% of the participants were male. The modified Union for International Cancer Control (mUICC) TNM stage was III in 1 patient (10%), IVa in 2 patients (20%), and IVb in 7 patients (70%). The Barcelona Clinic Liver Cancer (BCLC) stage was B in 1 patient (10%) and C in 9 patients (90%). Most patients (70%) had well-preserved liver function (Child–Pugh class A).

### 2.2. Overexpression of Plasma ADAM9 mRNA in Untreated HCC Patients

The plasma levels of *ADAM9* mRNA were tested in the five newly diagnosed and treatment-naïve HCC patients (Table 1, Subject #1 to #5) and expressed as fold changes compared to the healthy control group (*n* = 5, 100% female, mean age 34.2 years). The mean value of pre-treatment plasma *ADAM9* mRNA levels in the HCC patients was significantly higher than that in the healthy controls (3.001 ± 0.279 vs. 1.00 ± 0.005, *p* < 0.05) (Figure 1).

### 2.3. Decreased ADAM9 mRNA Expression Correlated with Response to Nivolumab

In four patients who were to receive nivolumab as second- or third-line treatment (Table 1, Subject #7 to #10), *ADAM9* mRNA expression in serum was significantly elevated compared to that in the healthy controls (Figure 2; pre-nivolumab mean, 323.39 ± 88.67 vs. 1.00 ± 0.000003, *p* < 0.05). To investigate the functional relevance between the change in *ADAM9* mRNA expression and clinical response, the serum levels of *ADAM9* mRNA were followed during the nivolumab therapy.

After three and four cycles of nivolumab therapy, respectively, Subjects #7 and #8 showed progressive disease (PD) on follow-up computed tomography (CT) scans. In these patients, serum levels of *ADAM9* mRNA in the follow-up were not significantly different from the pre-treatment levels (*p* > 0.05) (Figure 2A,B). Both patients succumbed to HCC progression and liver failure within 6 months since the start of nivolumab therapy.

In contrast, Subjects #9 and #10 started to reveal tumor regression after four cycles of nivolumab therapy. Their clinical courses are presented in Appendix A. Later, both patients exhibited partial response (PR) and survived longer than 6 months after the nivolumab therapy. In these patients, serum levels of *ADAM9* mRNA decreased significantly. In Subject #9, the serum *ADAM9* mRNA dropped from the pre-treatment level of 573.98 ± 5.16 to 523.85 ± 7.07 (*p* < 0.05) after two cycles, and further dropped to 262.58 ± 20.13 (*p* < 0.05) after 4 cycles of nivolumab therapy (Figure 2C). In Subject #10, it dropped from 323.88 ± 10.67 to 85.52 ± 5.59 (*p* < 0.05, Figure 2D) after two cycles of nivolumab therapy. 

### 2.4. Immunophenotype Changes following Nivolumab Therapy

Lymphocyte immunophenotypes were tested before and after three cycles of nivolumab therapy for Subject #7 and four cycles for the rest (Table 2 and Appendix A). Before nivolumab therapy, NK cells in Subjects #8 and #9 and cytotoxic T cells in Subject #9 were depleted, and these were therefore not detected for inhibitory checkpoint markers. In subject #8 and #10, PD-1 or T cell immunoglobulin- and mucin-domain-containing molecule-3 (TIM-3) positive cytotoxic T cells decreased significantly, but this change was not correlated with response to nivolumab. In contrast, TIM-3 positive helper T cells decreased in responders, but increased in non-responders.

### 2.5. Serum ADAM9 mRNA Expression was Completely Suppressed in Complete Response of HCC

Plasma *ADAM9* mRNA level was completely suppressed in one patient (Table 1 and Figure 1, Subject #6), who achieved complete response (CR). The clinical treatment course and the outcomes are detailed in Appendix A. This patient was a 59-year-old man with CHB HCC and had tumor recurrence after surgical resection. Since the HCC progressed on sorafenib and radiotherapy, nivolumab was administered in combination with activated autologous NK cell therapy (2–6 × 10^9^ cells/100 mL, intravenous infusion every 4 weeks). With serum AFP level increasing continuously, disease progression at the molecular level was suspected that nivolumab was switched to the third-line chemotherapy, regorafenib. To allow synergistic action of NK cells upon suppression of *ADAM9* protease and sMICA by regorafenib [24], the patient continued to receive the NK cell therapy every 4 weeks up to six times and then every 8 weeks up to a total of 15 times. At 6 months after beginning regorafenib with concomitant NK cell therapy, the patient achieved nearly complete regression of HCC. At this point, we acquired the patient’s blood sample for analysis and found that *ADAM9* mRNA was not detectable at all in his plasma (Figure 1). From then on, immunophenotype changes were also checked and a decrease in inhibitory checkpoint molecules was observed (Appendix A).

### 2.6. ADAM9 was Associated with HCC Prognosis in TCGA Database

To evaluate the effect of *ADAM9* expression on HCC prognosis, we performed *in-silico* analyses of 370 HCC patients from the TCGA database. Kaplan-Meier plots revealed that the group with *ADAM9* expression the higher than the median had a significantly poorer overall survival rate (Log-rank test *p* = 3.9 × 10^−4^) (Figure 3A). In addition, *ADAM9* was significantly upregulated in primary tumor tissues of HCC (*n* = 370) compared with adjacent normal liver tissues (*n* = 50) (*t*-test *p* = 4.6 × 10^−6^) (Figure 3B). Unlike *ADAM9*, other ADAM family genes *ADAM10* and *ADAM17* neither differed in their expression levels between HCC tumor tissues and adjacent normal liver tissues, nor showed significant correlation with survival analysis (Appendix A).

### 2.7. ADAM9 Expression is Positively Correlated with PD-1, TIM-3 and BTLA

The correlation between *ADAM9* expression and expression of immune checkpoint molecules (*PD-1, TIM-3, lymphocyte activation gene-3 (LAG-3)* and *B and T lymphocyte attenuator* (*BTLA))* in HCC patients (*n* = 370) from TCGA database was analyzed. *ADAM9* expression was positively correlated with the expression of *PD-1, TIM-3,* and *BTLA* but not with that of *LAG-3* (Figure 4). *TIM-3* had the strongest positive correlation with *ADAM9* (Correlation coefficient *r* = 0.37 and *p* = 1.3 × 10^−13^) (Figure 4B).

## 3. Discussion

In the present study, we found that the *ADAM9* blood mRNA level was significantly elevated in HCC patients compared to healthy controls. Furthermore, the magnitude of this elevation was much greater in the patients with previous treatment failure than in the newly diagnosed treatment-naïve patients. These findings suggested that *ADAM9* expression increase as HCC progresses, and that the immune evasion mechanisms involving *ADAM9* protease probably aggravate as HCC progresses. In addition, serum *ADAM9* mRNA levels decreased significantly in the two patients (one CHB and one non-viral HCC) who responded to nivolumab. Furthermore, *ADAM9* mRNA was not detected at all in the plasma of one HCC patient who achieved CR with regorafenib and NK cell combination immunotherapy. As such, the present study demonstrated, for the first time, the functional relevance of blood *ADAM9* mRNA levels with clinical response to HCC treatments. Since the sample size was too small in our study, we probed the TCGA data to find supporting evidence and found that higher *ADAM9* expression level was significantly associated with poor prognosis of HCC. In contrast, *ADAM10* and *ADAM17*, which were also reportedly related to MICA shedding [26], showed no such association. 

In our study, the change of serum *ADAM9* mRNA was easily correlated with the clinical response to nivolumab, while the observed lymphocyte immunophenotype changes were intriguing but not enough to draw a solid conclusion. Namely, Subject #8 did show some favorable changes in her lymphocyte immunophenotypes (mainly PD-1+ or TIM-3+ cytotoxic T cells) but tumor progression was still seen. Another interesting finding was the change of TIM-3 positive helper T cell; responders (Subjects #9 and #10) showed a decrease while non-responders (Subjects #7 and #8) showed an increase. In addition, *ADAM9* was strongly correlated with *TIM-3* in the TCGA database. These findings altogether suggest that key features of lymphocyte immunophenotype changes that can predict treatment response early on may exist. Our study indicated that the proportion of TIM-3 positive helper T cells may be a good candidate marker, and that unleashing helper T cell from exhaustion may be more important than unleashing cytotoxic T cells. Future studies are needed to elucidate the interplay between the ADAM9-MICA-NKG2D system and lymphocyte immunophenotypes, and to find the relevance between such factors and clinical outcome.

ADAMs belong to the zinc protease superfamily, and they are usually transmembrane proteins [27]. Containing disintegrin and metalloprotease domains, ADAMs take part in multiple cellular functions including cell adhesion and migration, proteolysis of the extracellular matrix and shedding of membrane proteins [27,28]. Several studies have indicated that ADAMs are involved in tumor development and progression of HCC [23,24,27,28,29,30,31,32,33,34,35,36,37,38,39,40]. Though the pathogenesis of HCC is multifactorial, it is largely due to hepatitis B (HBV) or C virus (HCV) infection and alcoholic or non-alcoholic fatty liver disease. These underlying liver conditions result in chronic inflammation and liver fibrosis, causing continuous remodeling of the extracellular matrix [27]. ADAMs are involved in this inflammatory process that leads to development of HCC [27]. Among several ADAMs associated with HCC, those related to MICA shedding are noteworthy as MICA is a critical part of cytotoxic cellular immunity. Our study revealed that only *ADAM9* was significantly associated with prognosis of HCC while others, *ADAM10* and *ADAM17*, were not. Regarding *ADAM9*, previous studies demonstrated that transcriptional suppression of *ADAM9* led to inhibition of proliferation and invasion activities of HCC cell lines [23,24,36,37,38,39,40]. Inhibition of *ADAM9* protease also showed similar results [41]. Some of these results were backed by the increased mMICA expression and subsequently increased susceptibility of HCC cells to NK cells [23,24,40]. On the other hand, treatment with interleukin (IL)-1β on HCC cell lines increased the expression of *ADAM9* and sMICA, and the IL-1β-treated HCC cells became more resistant to the cytolytic activity of NK cells [42].

MICA is a ligand of NKG2D, and it triggers NK cell or CD8+ T cell-mediated cytokine release and cytotoxicity towards the target cells [24]. MICA expression is induced in response to various types of stress such as heat, DNA damage, and viral infection [43]. The *ADAM9* protease that interferes with the MICA-NKG2D system was overexpressed in human HCC tissues [24]. In vitro, *ADAM9* knockdown increased mMICA expression, decreased sMICA production, and increased the cytolytic activity of NK cells against HCC cells [23]. Sorafenib suppresses the expression of ADAM9. Therefore, sorafenib treatment in vitro showed the same results as the *ADAM9* knockdown [23]. This important phenomenon was reproduced by another group. This time regorafenib was tested. Regorafenib suppressed the expression of *ADAM9* and ADAM10 to a greater extent than sorafenib [24]. This suppression led to mMICA accumulation and inhibition of sMICA production [24]. The authors suggested that regorafenib is superior to sorafenib as regorafenib suppresses not only *ADAM9* but also ADAM10 [24].

Beyond these experimental findings, the involvement of the MICA-NKG2D system was described in HCC patients. Both mMICA and sMICA showed elevated expression in human HCC [44,45,46]. Serum levels of sMICA were significantly elevated in chronic liver disease and HCC patients compared to healthy controls [44]. Furthermore, there was a stepwise increase in the level of sMICA as liver disease progressed from chronic hepatitis to cirrhosis, low-grade HCC and high-grade HCC [44]. In CHC patients, serum IL-1β levels were positively correlated with serum sMICA level, and serum IL-1β levels were significantly higher in CHC patients with HCC than those without HCC [42]. On the other hand, sMICA decrease could suggest favorable response to HCC treatment. For example, serum sMICA levels decreased, and NKG2D expression on NK cells and CD8+ T cells increased significantly after transcatheter arterial embolization therapy [44]. This could be related to our finding that *ADAM9* mRNA decreased in HCC patients who responded to treatments. Interestingly, there were divergent findings on the level of sMICA and its association with incidence or prognosis of HCC. In HCC with CHB, higher sMICA was correlated with vascular invasion and poor prognosis and was associated with a *G* allele of single nucleotide polymorphism (SNP) *rs2596542*, a risk allele for HBV-induced HCC (*p* = 0.029, odds ratio = 1.19) [47]. In contrast, *A* allele of SNP *rs2596542* was significantly associated with the higher risk of HCV-induced HCC (*p* = 4.21 × 10^−13^, odds ratio = 1.39) as well as low levels of sMICA [48]. These findings seemingly contradict each other, but could be explained by the fact that there may be an interplay between HCC etiology, i.e., the type of hepatitis virus or genetic factors and ADAM9-MICA-NKG2D system.

HCC occurrence in HBV infection can be partly ascribed to the perturbation of signaling pathways by HBV-encoded X protein (HBx) incorporated into the human genome [49]. HBx was associated with enhanced expression of MMPs such as *ADAM9* and ADAM10 [50,51]. More MMPs expressed could mean more MICA shedding. Therefore, this viral factor explains the association between the higher sMICA levels and the poorer prognosis of HCC with CHB [43,47]. In the absence of such interactions between virus and MMPs, the level of sMICA may, more or less, linearly follow that of mMICA expression, as mMICA is the source of sMICA [48]. In this case, low sMICA levels reflect low mMICA expression resulted from host genetic factor, which may indicate weakened cytotoxic immune surveillance, as was observed in HCV-induced HCC [48]. In addition, hepatitis B surface antigen is known to inhibit MICA expression via induction of cellular micro RNAs [52].

Anti-viral treatment for HBV and HCV was much improved by the development of new antiviral agents. This notwithstanding, the development of HCC still remains a concern even after HBV suppression [53,54] or HCV eradication [55]. In these patients, NK cells were usually observed to have depressed function with decreased capacity of interferon-γ production, impaired IL-15 production, or decreased expression of NK cell activation receptors including NKG2D [51,56,57]. Representing 30–50% of all hepatic lymphocytes, NK cells increase up to 90% in patients with hepatic malignancy. Therefore, ADAM9-MICA-NKG2D system may also serve as a good target for HCC prevention strategy in patients chronically infected with HBV or HCV. In this regard, it is encouraging that one group found an approved drug for anti-alcoholism, disulfiram, and showed that it effectively restored mMICA expression by inhibiting ADAM10 and did not have unfavorable off-target effects [58].

One of the unmet needs in this era of immunologic treatments for HCC or other cancers is a biomarker that can predict treatment response well in advance. It would be very useful if the marker could dictate the best cancer immunotherapy course. Despite the proven efficacy of nivolumab in HCC, expression of a known immune marker, PD-L1, in tumor tissues, fell short of predicting treatment response [16]. Our and previous studies have suggested that *ADAM9* mRNA may fill this unmet need [23,24,43]. *ADAM9* transcript elevation might suggest the existence of an *ADAM9* associated immune evasion mechanism in tumors. Decrease or increase of *ADAM9* mRNA after 2–4 cycles of treatment may help predict treatment response or failure early on. Finally, a decrease of *ADAM9* mRNA may indicate a candidate who can expect synergistic effects by adding adoptive cell therapy. Adoptive cell therapy composed of NK or CD8+ T cells have shown efficacy in treatment of HCC [3,17,22,59,60,61]. However, the patient population who had survival benefits was mostly restricted to those with minimal tumor burden after treatments with curative modality [17,22,62]. Therefore, there remains significant room for improvement of the adoptive cell therapy in patients with high burden of HCC [17,22,62]. This combination strategy may greatly enhance the survival of advanced HCC patients, as was shown in Subject #6. Thus, *ADAM9* mRNA may help select the patients who may benefit from such combination immunotherapy.

There were limitations in our study. The population size was too small. We did not investigate whether the findings of our study could be applied for other agents approved for advanced HCC or not. To complement such weaknesses, we probed the TCGA data and retrieved encouraging results. This pilot study demonstrates that *ADAM9* mRNA is associated with clinical response to HCC treatment and that there is an important link between the MICA-NKG2D system and prognosis of HCC, by showing how *ADAM9* mRNA expression changes over time as HCC patients received nivolumab therapy. Thus, *ADAM9* mRNA has potential as a biomarker to predict the clinical response of HCC patients, and the ADAM9-MICA-NKG2D system may be a good therapeutic target in HCC immunotherapy. Restoration of MICA-NKG2D signaling by suppressing *ADAM9* mRNA and its influence on the tumor microenvironment and its potential as a prognostic marker of human HCC patients should be investigated in future studies. The knowledge garnered will help us to formulate better strategies to manage the advanced HCC patients.

## 4. Materials and Methods

### 4.1. Patients

This study was conducted in CHA Bundang Hospital between January 2017 and October 2019. HCC patients who were to be treated with sorafenib, regorafenib or nivolumab as the standard-of-care therapy were eligible for this study. Patients who were 19 years of age or older and were willing to participate in this study were enrolled. For enrollment, the patient must have met the inclusion criteria which included presentation of HCC, diagnosed histologically or radiologically, in accordance with the guidelines of the American Association for the Study of Liver Diseases or the European Association for the Study of the Liver [5,10]. Other inclusion criteria were Child-Pugh score ≤ 7, BCLC stage B or C, Eastern Cooperative Oncology Group (ECOG) performance status score of 0 or 1, adequate bone marrow (hemoglobin ≥ 9 g/dL, granulocyte count >1000/mm^3^ and platelet count ≥ 40,000/mm^3^), adequate liver function (serum aspartate aminotransferase and alanine aminotransferase < 5 times of upper normal limit, bilirubin ≤ 3 mg/dL, prothrombin time international normalized ratio (PT INR) < 1.5) and adequate renal function (serum creatinine < 1.5 times the upper normal limit). In addition, HCC lesions must be measurable by CT or magnetic resonance imaging (MRI) and there should be at least one lesion ≥ 1.0 cm in maximum diameter. Exclusion criteria were evidence of malignant tumor of other than HCC, uncontrolled ascites, hepatic encephalopathy, major bleeding event within 30 days, severe bacterial infection, infection with HIV, major cardiac disease, anticoagulation therapy, pregnant or breast-feeding woman, dysphagia precluding drug administration, and other contraindications for systemic HCC therapy. Both written and oral consent was obtained prior to sample collection. The study was approved by the institutional review board (IRB) of the CHA Bundang Medical Center (IRB protocol: 2016-03-039-019).

### 4.2. Study Design and Protocol

This study was a prospective observational clinical research study. After enrollment, the first blood samples were drawn within 3 weeks before and 4 days after the commencement of chemotherapy with sorafenib, regorafenib or nivolumab. Afterwards, follow-up blood samples were acquired at the 5th, 10th and 20th week or when chemotherapy was terminated. Blood samples were drawn into two plain bottles (BD Vacutainer^®^, 5 mL), two EDTA bottles (BD Vacutainer^®^, 3 mL) and one heparin bottle (BD Vacutainer^®^, 10 mL). Immune cells acquired from the EDTA and heparin bottles were immediately subjected to examination of immunophenotypes and functional assays. The plasma samples in the heparin bottles and serum samples in the plain bottles were stored at −80 °C for further *ADAM9* mRNA quantification.

Patients’ age, sex, etiology of HCC and Child–Pugh score were recorded. Clinical and TNM stages were classified according to the BCLC clinical stage and the mUICC stages, respectively [5,63]. Tumor response (CR, PR, stable disease, and PD) was evaluated in accordance with the modified Response Evaluation Criteria in Solid Tumors (mRECIST) [64] every 10–12 weeks for more than 1 year via dynamic liver CT or MRI. Complete blood cell count, blood chemistry (liver function, renal function, metabolic function, electrolyte), PT INR, and tumor markers (AFP and PIVKA-II) were regularly checked as part of standard care.

### 4.3. mRNA Isolation and Real-Time PCR

To evaluate the serum or plasma *ADAM9* mRNA expression levels, quantitative real-time reverse transcription-polymerase chain reaction (RT-PCR) was performed using a CFX Connect real-time system (Bio-Rad, Hercules, CA, USA). Total RNA was isolated from serum or plasma samples using the miRNeasy Serum/Plasma Kit (Qiagen, Hilden, Germany). For normal controls, we used banking serum and plasma from women (*n* = 5, mean age 34.2 years, range 29–41 years) who delivered at term (≥35 gestational weeks) because we had difficulty with collection of normal serum. The collection and use of these samples for research purposes was approved by the IRB of CHA Hospital (Seoul, Korea) (IRB protocol: 2006-12). The mRNA levels of *ADAM9* and glyceraldehyde phosphate dehydrogenase (GAPDH) were determined by quantitative real-time RT-PCR using SYBR green mastermix (Roche, Basel, Switzerland) according to the manufacturer’s instructions. PCR reactions were performed by denaturation at 95 °C for 10 min followed by 45 cycles of amplification at 95 °C for 10 s, 55 °C for 15 s, and then melting curves were performed after PCR amplification under the following conditions: 60 °C to 95 °C with a temperature transition rate of 1 °C/s. The sequence of the PCR primers used for detection of *ADAM9* and *GAPDH* were as follows: *ADAM9* forward primer, 5′-GGAAACTGCCTT CTTAATATTCCAAA-3′, *ADAM9* reverse primer, 5′-CCCAGCGTC CACCAACTTAT-3′, *GAPDH* forward primer, 5′- CTCCTCTTCGGCAGCACA-3′, *GAPDH* reverse primer, 5′- AACGCTTCACCTA ATTTG CG T -3′. *GAPDH* mRNA from each sample was quantified as an endogenous control of internal RNA. All experiments were performed in duplicate.

### 4.4. Flow Cytometric Analysis

HCC patient’s blood in the heparin bottle was transferred to a 15 mL conical tube containing 5 mL filcol-paque plus (GE Healthcare, Chicago, IL, USA) and centrifuged at 2000 rpm for 10 min. Plasma was transferred to a 5 mL cryovial (Corning Inc, Corning, NY, USA) and stored at −80 ℃. The buffy coat layer was transferred to a new 15 mL conical tube washed 2 times with PBS. Peripheral blood mononuclear cells (PBMCs) were stained with anti-CD3-eFluor 506 (ebioscience, San Diego, CA, USA), anti-CD3-eFluor 450 (ebioscience, USA), anti-CD4-eFluor 450 (ebioscience, USA), anti-CD4-PE (ebioscience), anti-CD8-Alexa Fluor 700 (ebioscience), anti-CD19-PE (ebioscience), anti-CD56-Alexa Fluor 700 (ebioscience), anti-CD16-PerCP/Cy5.5 (BioLegend, San Diego, CA, USA), anti-NKG2D-FITC (ebioscience), anti-CD158b-PerCP/Cy5.5 (BioLegend), anti-TIM3-FITC (ebioscience), anti-LAG-3-PE (BioLegend), anti-BTLA-APC/Cy7 (BioLegend), anti-PD-1-APC (BioLegend), anti-CTLA4-APC (Miltenyi Biotec, Bergisch Gladbach, Germany), anti-mouse IgG1 kappa isotype control-FITC (ebioscience, USA), anti-mouse IgG1 kappa isotype control-PE (ebioscience), anti-mouse IgG1 kappa isotype control-PerCP/Cy5.5 (BioLegend), REA control-APC (Miltenyi Biotec), anti-mouse IgG1 kappa isotype control-Alexa Fluor 700 (ebioscience), anti-mouse IgG1 kappa isotype control-APC/Cy7 (BioLegend), anti-mouse IgG1 kappa isotype control-eFluor 506 (ebioscience), anti-mouse IgG1 kappa isotype control-eFluor 450 (ebioscience). Stained cells were analyzed with CytoFLEX flow cytometry (Beckman Coulter, Brea, CA, USA) and resulting data were analyzed using Kaluza version 1.5a analysis software (Beckman Coulter).

### 4.5. Statistical Analysis

Data are expressed as the mean (± standard deviation) or frequencies (percentages), as appropriate. The statistical significance of differences between the groups was determined by Student *t* test or two-sample *t* test. A *p*-value of <0.05 was considered statistically significant. Statistical analyses were performed with SPSS 22.0 (SPSS Inc., Chicago, IL, USA).

### 4.6. In-Silico Analysis with TCGA Database

We downloaded transcriptomic, survival and clinical data of HCC patients (indexed as LIHC) from the Xena TCGA database hub (https://xenabrowser.net). The transcriptomic data included 370 patients and was generated by the University of North Carolina TCGA genome characterization center. The survival data includes information on overall survival. Statistical analyses were performed with *t*-test, Pearson’s correlation analysis, Cox-regression and log-rank analysis. All statistical analyses with TCGA dataset were performed with Python (Version 2.7.10) and R-studio (Version 1.1.456).

## 5. Conclusions

*ADAM9* mRNA was overexpressed in blood samples of patients with advanced HCC. Decreased levels of *ADAM9* mRNA in the blood was significantly associated with clinical response to HCC treatment with nivolumab. Also, using the TCGA database, we found that higher expression of *ADAM9* in HCC tumor tissues is associated with poor survival of HCC patients. Therefore, *ADAM9* mRNA has a potential as a biomarker predicting clinical response, and the ADAM9-MICA-NKG2D system may serve as a therapeutic target of HCC immunotherapy.

## Figures and Tables

**Figure 1 cancers-12-00745-f001:**
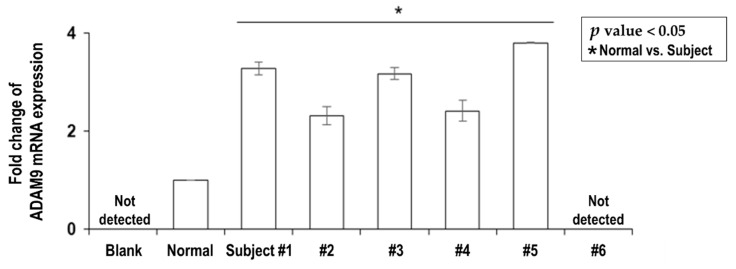
Expression level of *ADAM9* mRNA in plasma samples of treatment naïve advanced HCC patients and one HCC patient with near complete remission. Each patient (Subject #1 to #5) with newly diagnosed and treatment naïve HCC had significantly elevated expression of *ADAM9* mRNA in plasma compared with normal controls (3.001 ± 0.279 vs. 1.00 ± 0.005, *p* < 0.05). The HCC patient who had reached near complete remission (Subject #6) did not express *ADAM9* mRNA in plasma at all.

**Figure 2 cancers-12-00745-f002:**
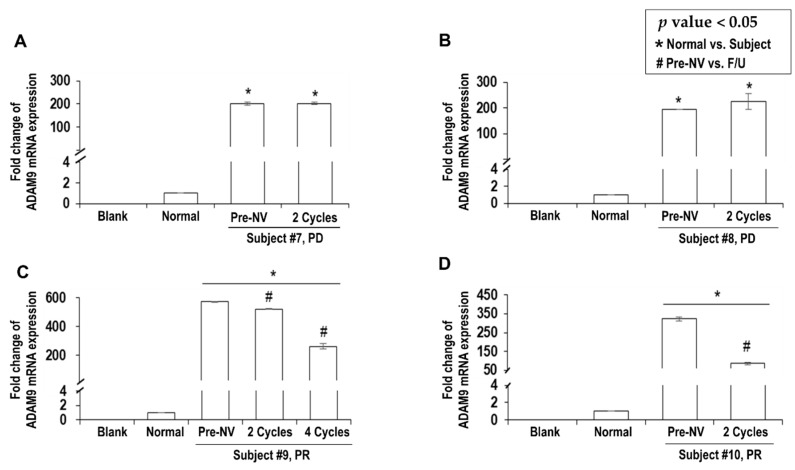
Expression level changes of serum *ADAM9* mRNA in advanced HCC patients treated with nivolumab. The number of nivolumab cycles completed prior to follow-up blood sampling is denoted on the *X*-axis. After 2 cycles of nivolumab therapy, non-responders did not show significant change in *ADAM9* mRNA (**A**, Subject #7; **B**, Subject #8). In contrast, responders exhibited a significant decrease in *ADAM9* mRNA (**C**, Subject #9, from 573.98 ± 5.16 to 523.85 ± 7.0 (*p* < 0.05) after 2 cycles, and further down to 262.58 ± 20.13 (*p* < 0.05) after 4 cycles; **D**, Subject #10, from 323.88 ± 10.67 to 85.52 ± 5.59 (*p* < 0.05) after 2 cycles). Abbreviations: NV, nivolumab; F/U, follow-up; PD, progressive disease; PR, partial response.

**Figure 3 cancers-12-00745-f003:**
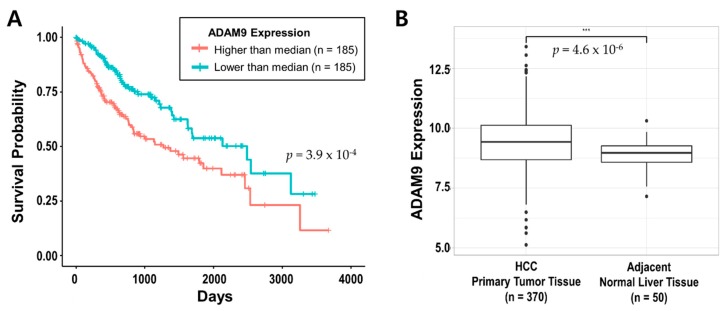
Effect of *ADAM9* expression on HCC prognosis in the TCGA database. (**A**) Kaplan–Meier plot of HCC patients (*n* = 370) according to *ADAM9* expressions level higher or lower than median (*n* = 185 for each group). (**B**) Box-plot comparing *ADAM9* expression between HCC primary tumor (*n* = 370) and adjacent normal liver tissue (*n* = 50). Abbreviations: HCC, hepatocellular carcinoma; TCGA, The Cancer Genome Atlas.

**Figure 4 cancers-12-00745-f004:**
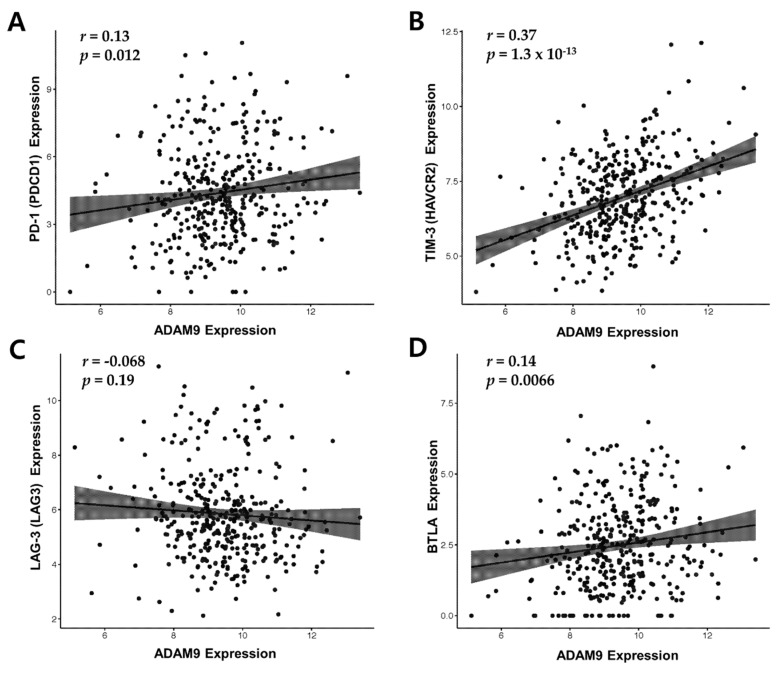
Correlation between *ADAM9* expression and four immune checkpoint genes *PD-1* (**A**), *TIM-3* (**B**), *LAG-3* (**C**), and *BTLA* (**D**). Abbreviations: *r*, Pearson’s correlation coefficient; PD-1, programmed cell death 1; TIM-3, T cell immunoglobulin- and mucin-domain-containing molecule 3; LAG-3, lymphocyte activation gene 3; BTLA, B and T lymphocyte attenuator.

**Table 1 cancers-12-00745-t001:** Demographic details and clinical characteristics of study participants (*n* = 10).

Subject No.	Age	Sex	Etiology	Antiviral Therapy	Serum AFP (ng/mL)	Child-Pugh Class	Tumor Size (cm)	PV Invasion (Vp) *	mUICC Stage	Extra-Hepatic Metastasis	Type of Therapy
#1	58	M	HBV	TDF	97,387	A	8	4	IVb	Yes	Sorafenib
#2	61	M	HBV	ETV	>200,000	B	22	2	Ivb	Yes	Sorafenib
#3	61	M	HBV	ETV	190	B	5.2	3	Ivb	Yes	Sorafenib
#4	55	M	HBV	TDF	71.7	A	3	0	III	No	TACE, Sorafenib
#5	58	M	HBV	ETV	14.7	A	10	2	Ivb	Yes	Sorafenib, Nivolumab
#6	59	M	HBV	ETV	82.4	A	4	2	Iva	No	Sorafenib, Nivolumab + NK cell therapy, Regorafenib + NK cell therapy
#7	45	M	HBV	TDF	154.7	B	11	4	Iva	No	Sorafenib, Nivolumab
#8	44	F	HCV	DAC/SUN	6519.4	A	4.5	0	Ivb	Yes	Sorafenib, Regorafenib, Nivolumab
#9	58	F	HBV	TDF	66	A	9	2	Ivb	Yes	Sorafenib, Nivolumab Regorafenib
#10	76	M	NASH	none	4594.1	A	5.2	1	Ivb	Yes	Sorafenib, Nivolumab

Abbreviations: No, number; M, male; F, female; AFP, alpha-fetoprotein; PV, portal vein; mUICC, modified Union for International Cancer Control; HBV, hepatitis B virus; HCV, hepatitis C virus; NASH, nonalcoholic steatohepatitis; TACE, transarterial chemoembolization; NK, natural killer; TDF, tenofovir disoproxil fumarate; ETV, entecavir; DAC, daclatasvir; SUN, asunaprevir. * The extent of portal vein invasion (Vp) by tumor thrombosis was documented according to the Liver Cancer Study Group of Japan classification: Vp0 = no portal vein invasion, Vp1 = segmental portal vein invasion, Vp2 = right anterior/posterior portal vein, Vp3 = right/left portal vein and Vp4 = main trunk [25].

**Table 2 cancers-12-00745-t002:** Lymphocyte immunophenotypes of the 4 patients who received nivolumab therapy (%).

Cell Type	Phenotype Marker	Non-Responders				Responders			
		Subject #7		Subject #8		Subject #9		Subject #10	
		Pre-NV	3 Cycles	Pre-NV	4 Cycles	Pre-NV	4 Cycles	Pre-NV	4 Cycles
T cells	CD3^+^	63.62	52.98	74.53	84.57	6.18	19.02	45.01	53.65
Helper T cells	CD3^+^CD4^+^	27.46	26.36	59.91	52.43	5.51	10.58	30.57	35.58
Cytotoxic T cells	CD3^+^CD8^+^	34.15	20.03	13.95	30.7	0.73	7.41	14.57	18.75
B cells	CD19^+^	10.15	7.27	1.88	1.06	3.21	60.7	19.1	5.84
NK cells	CD3^−^CD56^+^	10.77	5	0.21	5.52	0.61	7.49	19.87	18.88
NKT cells	CD3^+^CD56^+^	4	2.37	2.03	2.59	0.23	0.78	6.68	4.63
Cytotoxic T cells	CD3^+^CD8^+^PD-1^+^	3.95	6.48	22.34	2.87	ND	0.12	24.65	7.99
	CD3^+^CD8^+^TIM3^+^	21.05	52.11	15.06	5.3	ND	45.32	11.57	6.73
	CD3^+^CD8^+^LAG3^+^	6.58	16.47	38.57	37.64	ND	3.33	29.95	30.63
	CD3^+^CD8^+^BTLA^+^	14.47	11.32	29.87	54.32	ND	6.7	2.7	1.79
Helper T cells	CD3^+^CD4^+^PD-1^+^	0	1.22	15.44	2.15	7.69	7.34	13.09	3.3
	CD3^+^CD4^+^TIM3^+^	19.81	37.3	3.42	11.89	69.74	45.52	10.2	4.01
	CD3^+^CD4^+^LAG3^+^	0.94	16.51	35.16	25.04	8.21	2.42	6.93	4.53
	CD3^+^CD4^+^BTLA^+^	13.16	9.4	24.81	29.19	0.51	2	1.94	11.14
NK cells	CD3^−^CD56^+^PD-1^+^	8.7	1.24	ND	1.58	ND	1.19	4.39	3.94
	CD3^−^CD56^+^TIM-3^+^	4.35	22.39	ND	21.45	ND	33.6	25.45	14.6
	CD3^−^CD56^+^LAG3^+^	0	2.74	ND	14.51	ND	5.06	46.1	25.15
	CD3^−^CD56^+^BTLA^+^	47.83	26.18	ND	9.15	ND	3.45	1.63	1.47

Blood samples for the follow-up were acquired after three cycles of nivolumab therapy for Subject #7 and four cycles for the rest. Abbreviations: NV, nivolumab; HCC, hepatocellular carcinoma; NK, natural killer; NKT, natural killer-T; CD, cluster differentiation; ND, not detected; PD-1, programmed cell death 1; TIM-3, T cell immunoglobulin- and mucin-domain-containing molecule 3; LAG-3, lymphocyte activation gene 3; BTLA, B and T lymphocyte attenuator.

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
