# Peer review of "A Disintegrin and Metalloproteinase 9 (ADAM9) in Advanced Hepatocellular Carcinoma and Their Role as a Biomarker During Hepatocellular Carcinoma Immunotherapy"

_cancers, 2020, doi:10.3390/cancers12030745_

Round 1

Reviewer 1 Report

The manuscript “A Disintegrin and Metalloproteinase 9 (ADMA9) in Advanced Hepatocellular Carcinoma (HCC) and it’s role as a Biomarker during HCC immunotherapy” reports an observational study aimed at investigating the role of ADAM9 as a biomarker of immunotherapy for HCC. The authors have studied and compared the mRNA levels of ADAM9 as well as the levels of cd8+ T and cd56+ NK cells in patients at different stages of treatment. Although the results are of interest to both immunotherapy and HCC fields, there are some limitations in the study which could add to the overall quality of the reported results.

Authors have rightly pointed out that the biggest limitation of the study is the number of patients. The involvement of ADAM9 in MICA shedding has been known for a while and molecular mechanisms have also been studied (e.g. by Kohga et al.) Novelty of this work lies on the possibility of using ADAM9 as a prognostic marker or even a therapeutic target. With data comparing only 2 responders (Subjects 9 and 10) with 2 non-responders (Subjects 7 and 8) and 5 naïve cases, it is a bit difficult to establish ADAM9 as a biomarker even though the evidence is compelling. I realize the practical constraints for getting samples but Increasing the number of patients will have a significant impact in overall HCC management. I strongly suggest the authors to probe public databases for HCC cohorts and check for the availability of sequencing data to compare the levels of ADAM9 in patients at different stages of treatment. This will be even more convincing. Other ADAM proteins (e.g. ADAM17) has been shown to have similar effect on MICA shedding. Did the authors look into this? Additionally, it might also be interesting to look into the expression levels of HCC-specific proteins like AFP or GPC3 and so forth. Would the authors be able to provide the status of CD8+ T cells and CD56+ NK cells for subjects 1-5? If not, it might be very interesting to see this in at least patient 6 who showed CR to compare with the two responders (patients 9 and 10). Why is Fig 3 and 4 from Subject 9 whereas Fig 5 from subject 10? Flow cytometry for LAG-3 and BTLA (Figure S1) is lacking. The authors should show flow results for both subjects 9 and 10 in figures 3, 4 and 5 if possible. Some mechanistic studies in HCC cell lines or in vivo would have been good. Maybe inhibition of ADAM9 in HCC mouse model to check OS.

Author Response

Reviewer 1

Reviewer) The manuscript “A Disintegrin and Metalloproteinase 9 (ADMA9) in Advanced Hepatocellular Carcinoma (HCC) and it’s role as a Biomarker during HCC immunotherapy” reports an observational study aimed at investigating the role of ADAM9 as a biomarker of immunotherapy for HCC. The authors have studied and compared the mRNA levels of ADAM9 as well as the levels of cd8+ T and cd56+ NK cells in patients at different stages of treatment. Although the results are of interest to both immunotherapy and HCC fields, there are some limitations in the study which could add to the overall quality of the reported results.

Authors have rightly pointed out that the biggest limitation of the study is the number of patients. The involvement of ADAM9 in MICA shedding has been known for a while and molecular mechanisms have also been studied (e.g. by Kohga et al.) Novelty of this work lies on the possibility of using ADAM9 as a prognostic marker or even a therapeutic target. With data comparing only 2 responders (Subjects 9 and 10) with 2 non-responders (Subjects 7 and 8) and 5 naïve cases, it is a bit difficult to establish ADAM9 as a biomarker even though the evidence is compelling. I realize the practical constraints for getting samples but Increasing the number of patients will have a significant impact in overall HCC management. I strongly suggest the authors to probe public databases for HCC cohorts and check for the availability of sequencing data to compare the levels of ADAM9 in patients at different stages of treatment. This will be even more convincing.

Answer) Thank you so much your critical appraisal. We, the authors, fully agree with you. It is a pity that we could not enroll more patients due to the limitation of funding and time. To comply with your advice, we performed TCGA analyses. We hope the additional findings may come to your expectation.

Line 190-219 (Result Sections);

2.6. ADAM9 was Associated with HCC Prognosis in TCGA Database

2.7. ADAM9 Expression Positively Correlated with PD-1, TIM-3 and BTLA

Line 424-430 (Method Section);

4.6. In-silico analysis with TCGA database

Reviewer) Other ADAM proteins (e.g. ADAM17) has been shown to have similar effect on MICA shedding. Did the authors look into this?

Answer) Thank you so much for your insightful opinion. Yes, we have included ADAM17 and ADAM10 into our analyses with TCGA. The results are presented in Figure S9 and S10. The results are quite interesting and indicate that, out of all three ADAMs that are involved with MICA shedding in HCC, ADAM9 is the most important one that it is significantly associated with prognosis of HCC patients and correlated very strongly with TIM-3. It gives us much idea on direction of future studies. Thank you very much for your contribution.

Line 195-198 (Results 2.6);

“Unlike ADAM9, other ADAM family genes, including ADAM10 and ADAM17, did not differ in their expression levels between HCC tumor tissues and adjacent normal liver tissues, and neither showed significant correlation with survival analysis (Figure S9 and S10). “

Reviewer) Additionally, it might also be interesting to look into the expression levels of HCC-specific proteins like AFP or GPC3 and so forth.

Answer) Thank you so much for your opinion. Unfortunately, we could not look into AFP or GPC3 in this study, but we will consider those in our future studies. Thank you. 

Reviewer) Would the authors be able to provide the status of CD8+ T cells and CD56+ NK cells for subjects 1-5? If not, it might be very interesting to see this in at least patient 6 who showed CR to compare with the two responders (patients 9 and 10). Why is Fig 3 and 4 from Subject 9 whereas Fig 5 from subject 10? Flow cytometry for LAG-3 and BTLA (Figure S1) is lacking.

Answer) Thank you for your constructive advice. At first, we were not sure how to interpret the immunophenotype results that we had shown the results selectively. Following reviewers’ instruction, we have run more tests and included all the data of FACS results in Table 2 (Figure S3-S6) for Subject #7-#10 and Table S2 for Subject #6. In doing so, we could also see the meaning of the immunophenotype changes. Thank you very much for your contribution.

Line 162-167 (Results 2.4.);

“Lymphocyte immunophenotypes were tested before and after 3 cycles of nivolumab therapy for Subject #7 and 4 cycles for the rest (Table 2 and Figure S3 to S6). Before nivolumab therapy, NK cells in Subject #8 and #9 and cytotoxic T cells in Subject #9 were depleted that those were not detected for inhibitory checkpoint markers. In subject #8 and #10, PD-1 or TIM3 positive cytotoxic T cells decreased significantly, but such change was not correlated with response to nivolumab. On the contrary, TIM3 positive helper T cells decreased in responders, but increased in non-responders .”

Line 185-188 (Results 2.5.)    

At this point, we acquired his blood sample for this study, and ADAM9 mRNA was not detected at all in his plasma (Figure 1). From then on, immunophenotype change was also checked, and decrease of inhibitory checkpoint molecules was observed (Table S2).

Line 246-255 (Discussion)

“In addition, change of serum ADAM9 mRNA was easily correlated with response to nivolumab while lymphocyte immunophenotype change was hard to interpret. This suggests that restoring anti-cancer immunity require more than just unleashing lymphocytes from exhaustion. For example, Subject #8 did have some favorable response in his lymphocyte immunophenotypes but progressed. A biggest distinction between Subject #8 (non-responder) and #10 (responder) was the change of TIM-3 positive helper T cell that Subject #8 failed to have it decreased. This may suggest that it be more important to unleash helper T cells than cytotoxic T cells to orchestrate various immune cells and tumor microenvironment altogether towards anti-cancer immunity. In addition, the strong correlation between ADAM9 and TIM-3 implies that successful inhibition TIM-3 may lead to suppression of ADAM9.”

Reviewer) The authors should show flow results for both subjects 9 and 10 in figures 3, 4 and 5 if possible.

Answer) Thank you. We added flow results of clinical course of subject #9 and #10 in Figure S1 and S2, respectively.

Reviewer) Some mechanistic studies in HCC cell lines or in vivo would have been good. Maybe inhibition of ADAM9 in HCC mouse model to check OS. 

Answer) Thank you so much for your constructive idea. Due to the time limitation, we could not do the in vitro or in vivo experiments. We will consider those in our future studies. Thank you.

Reviewer 2 Report

Dear Authors,

1. This manuscript should be considered as case report study than as original article.

2. Most of the described results concern only 4 patients, separately 1 patient.

3. Most of patients were infected with HBV. There is no data about their HBV therapy.

4. Please do not repeat the results (numbers) in the discussion.

5. In the current form, the discussion is a fragment of review, too few comments to the obtained results.

Author Response

Reviewer 2

Dear Authors,

  1. This manuscript should be considered as case report study than as original article.

Answer) Thank you for your comment. We, the authors, fully understand what is meant by your comment. We are very regretful about it. To compensate, we ran follow-up tests and did TCGA analyses during the past 2 weeks. We hope the added new result may help you think otherwise.  

Line 190-219 (Result Sections);

2.6. ADAM9 was Associated with HCC Prognosis in TCGA Database

2.7. ADAM9 Expression Positively Correlated with PD-1, TIM-3 and BTLA

Line 424-430 (Method Section);

4.6. In-silico analysis with TCGA database

  1. Most of the described results concern only 4 patients, separately 1 patient.

Answer) Thank you for your comment. We are very sorry about the limited number of study participants. We had difficulties in acquiring enough funding, so we could not include many participants for the follow-up laboratory measurements. Since we are constantly trying to get more fund, we hope to enroll more participants in the future. Thank you so much for your concern.

  1. Most of patients were infected with HBV. There is no data about their HBV therapy.

Answer) Thank you very much for your meticulous advice. We added the info in Table 1.

  1. Please do not repeat the results (numbers) in the discussion.

Answer) Thank you for your meticulous advice. It is corrected accordingly.

  1. In the current form, the discussion is a fragment of review, too few comments to the obtained results.

Answer) Thank you for your insightful opinion. As we have done more analyses during the revision, we could discuss more about the obtained results than before. We hope the change may help you think otherwise. Thank you for your understanding.  

Reviewer 3 Report

The study is very interesting, and I also think the system treatment on HCC might be a good choice for other kinds of tumors. However, the results are not so solid.

There are only about 10 patients studied in the manuscript, so the conclusion made based on the limited data might not be so strong. Did the author check the correlation based on TCGA or other liver cancer database? Many studies showed immune therapy is not as useful as it in other kinds of tumors, however, the author showed combination immunotherapy and sorafenib had a good outcome, could the author give me discussion, and discrete the molecular background of different HCC patients. From figure 2, patients 7 and 8 didn’t get a good response to nivolumab treatment, so did the author also test the immune response changes on these immune cells? In the 2.3 part, what’s the time point of test after nivolumab treatment did the author follow up the immune cells changes during all the treatment time? To figure 4 and 5, the author could combine them together to describe the data and interpret the results. Meanwhile, since they are form risk factors of liver cancer, maybe the author could find a potential reason for the different responses? And could the author also draw a figure similar to figure 6 to compare the complete response(#6 patient) and partial response patients( #9,#10 patient) to test the hypothesis?

In general, it’s a good discovery, but it needs more data and test to verify the conclusion.

Author Response

Reviewer) The study is very interesting, and I also think the system treatment on HCC might be a good choice for other kinds of tumors. However, the results are not so solid.

There are only about 10 patients studied in the manuscript, so the conclusion made based on the limited data might not be so strong. Did the author check the correlation based on TCGA or other liver cancer database?

Answer) Thank you for your insightful opinion. During the revision, yes, we looked into TCGA data and presented the results. We hope the results may come to your expectation.

Line 190-219 (Result Sections);

2.6. ADAM9 was Associated with HCC Prognosis in TCGA Database

2.7. ADAM9 Expression Positively Correlated with PD-1, TIM-3 and BTLA

Line 424-430 (Method Section);

4.6. In-silico analysis with TCGA database

Reviewer) Many studies showed immune therapy is not as useful as it in other kinds of tumors, however, the author showed combination immunotherapy and sorafenib had a good outcome, could the author give me discussion, and discrete the molecular background of different HCC patients. From figure 2, patients 7 and 8 didn’t get a good response to nivolumab treatment, so did the author also test the immune response changes on these immune cells?

Answer) Thank you for your meticulous remarks. Regarding the effect of combination immunotherapy with regorafenib and NK cell therapy in our Subject #6, we also hope to be able to give you discrete molecular background and discussions accordingly. However, he was enrolled too late for this study that we can only speculate the molecular mechanism based on previous studies by Kohga et al(Ref. 24) and Arai et al(Ref. 25). Instead, we ran follow-up FACS for patients #7-#8 and altogether presented in Table 2. The raw data are presented in Figures S3-S6. We also added the potential molecular reason of different responses. It seems that unleashing helper T cell is much more important that the critical difference between subject #8(non-responder) and subject #10(responder) was the TIM3+ helper T cell. In addition, TIM3+ and ADAM9 revealed strong positive correlation in our TCGA analyses, so we guess that helper T cell may have an important role to bring about cooperation between various immune cells and tumor microenvironment and subsequent restoration of anti-cancer immunity. Though it cannot be discussed conclusively, we think it is still meaningful to find the connection between immune checkpoints and ADAM9.     

Line 162-167 (Results 2.4.);

“Lymphocyte immunophenotypes were tested before and after 3 cycles of nivolumab therapy for Subject #7 and 4 cycles for the rest (Table 2 and Figure S3 to S6). Before nivolumab therapy, NK cells in Subject #8 and #9 and cytotoxic T cells in Subject #9 were depleted that those were not detected for inhibitory checkpoint markers. In subject #8 and #10, PD-1 or TIM3 positive cytotoxic T cells decreased significantly, but such change was not correlated with response to nivolumab. On the contrary, TIM3 positive helper T cells decreased in responders, but increased in non-responders.”

Line 246-254 (Discussion)

“In addition, change of serum ADAM9 mRNA was easily correlated with response to nivolumab while lymphocyte immunophenotype change was hard to interpret. This suggests that restoring anti-cancer immunity require more than just unleashing lymphocytes from exhaustion. For example, Subject #8 did have some favorable response in his lymphocyte immunophenotypes but progressed. One important distinction between Subject #8 (non-responder) and #10 (responder) was the change of TIM-3 positive helper T cell. This may suggest that it be more important to unleash helper T cells than cytotoxic T cells to orchestrate various immune cells and tumor microenvironment altogether. In addition, the strong correlation between ADAM9 and TIM-3 implies that successful inhibition TIM-3 may lead to suppression of ADAM9.”

Reviewer) In the 2.3 part, what’s the time point of test after nivolumab treatment did the author follow up the immune cells changes during all the treatment time?

Answer) Thank you for your meticulous remark. The info is provided as the following.

Line 162-163 (Results 2.4.): 

“Lymphocyte immunophenotypes were tested before and after 3 cycles of nivolumab therapy for Subject #7 and 4 cycles for the rest (Table 2 and Figure S3 to S6).”  

Reviewer) To figure 4 and 5, the author could combine them together to describe the data and interpret the results.

Answer) Thank you for your meticulous remark. We did accordingly in Table 2 and Figure S3-S6. Thank you so much.

Reviewer) Meanwhile, since they are form risk factors of liver cancer, maybe the author could find a potential reason for the different responses?

Answer) Thank you for your insightful opinion. We could not tell the risk factors, per se, for this study. However, we think the immunophenotype change gave us a meaningful result that helper T cell is much more important that the critical difference between subject #8(non-responder) and subject #10(responder) as was answered above.  

Reviewer) And could the author also draw a figure similar to figure 6 to compare the complete response(#6 patient) and partial response patients( #9,#10 patient) to test the hypothesis?

Answer) Thank you for your comment. Following your instruction and other reviewers, we added Figure S1 and S2 to show the clinical courses of subject #9 and #10.  

Reviewer) In general, it’s a good discovery, but it needs more data and test to verify the conclusion.

Answer) Thank you for your insightful remark. The newly added part on TCGA analysis may supplement in that regards. Thank you for understanding.  

Reviewer 4 Report

The paper is interesting thematically, but rather chaotically developed.
In the abstract, the keyword is, among others, regorafenib. Where are the results regarding its effect on surface antigens?
In the results (chapter 2.1), the authors write that patients were treated with sorafenib, regorafenib or nivolumab. The results only apply to nivolumab. If they were therapeutic failures, then it should be clearly stated in the material and methods chapter.
There is also no defined control group - how many people, average age, gender. After all, in figures 1 and 2 they are marked.
In figure 3B, the numerical values ​​should be placed on the left.
Should CD16 not be used except CD56 and CD3 ?.
Figure 5 is incorrectly signed. The graphs do not show expression, but the percentage of cells expressing the given antigen. Expression of the examined antigen should be given as MFI.
The results are based on too few patients. Despite the fact that advanced hepatocellular carcinoma is not a common cancer, I think that in almost 3 years of research more patients can be obtained. This is especially true for non-viral HCC patients. Such small groups prevent statistical analysis.
References have been selected and mostly developed in the last 5-10 years.

Author Response

Reviewer 4

Reviewer) The paper is interesting thematically, but rather chaotically developed.
In the abstract, the keyword is, among others, regorafenib. Where are the results regarding its effect on surface antigens?

Answer) Thank you for your insightful comment. We fell short to show them that we erased regorafenib from the keywords.

Reviewer) In the results (chapter 2.1), the authors write that patients were treated with sorafenib, regorafenib or nivolumab. The results only apply to nivolumab. If they were therapeutic failures, then it should be clearly stated in the material and methods chapter.

Answer) Thank you for your insightful comment. We simply could not follow them (Subject #1-#5) yet. We thought those patients are treatment naïve ones that pretreatment level of ADAM9 mRNA would suffice for the time being. We fell short for funding that we had certain limitation in including the all the patients for the follow-up laboratory measurements. We are constantly soliciting funding for this study. So, we may be able to measure samples from those patients in the future. 

Reviewer) There is also no defined control group - how many people, average age, gender. After all, in figures 1 and 2 they are marked.

Answer) Thank you for your meticulous remark. We added the info as the following.

Line 125 (Results 2.1.);

“normal healthy control group (n = 5, 100% female, mean age 34.2 years).”

Line 386-388 (Methods 4.3);

For normal controls, we used banking serum and plasma from women (n = 5, mean age 34.2 years, range 29-41 years) who delivered at term (≥35 gestational weeks) because we had difficulty with collection of normal serum

Reviewer) In figure 3B, the numerical values ​​should be placed on the left.
Should CD16 not be used except CD56 and CD3 ?.
Figure 5 is incorrectly signed.

Answer) Thank you so much. We rearranged the data in Table 2 and Figure S3-S6. CD16 was not meaningful that we decided not to show in the results except for Subject #6.

Reviewer) The graphs do not show expression, but the percentage of cells expressing the given antigen. Expression of the examined antigen should be given as MFI.

Answer) Thank you for your insightful comment. We feel regretful that we could not give the values in MFI.

Reviewer) The results are based on too few patients. Despite the fact that advanced hepatocellular carcinoma is not a common cancer, I think that in almost 3 years of research more patients can be obtained. This is especially true for non-viral HCC patients. Such small groups prevent statistical analysis.

Answer) Thank you for your practical comment. We had difficulties in planning and getting enough fund to actually run the study. In part, the small number of participants is also due to the limited fund that we have got. Nevertheless, we are constantly trying to get more fund that we will be able to include more participants in the future. Thank you for your understanding.  

Reviewer) References have been selected and mostly developed in the last 5-10 years.

Answer) Thank you for your insightful comment. We have researched all the relevant literature but found that the topic has not been vigorously studied in the recent 5 years. To our best knowledge, we have cited the most recent studies on the topic. Thank you so much for your concern.

Round 2

Reviewer 1 Report

It seems that the authors have added text in the manuscript in a hurry and sloppiness is obvious when comparing with the first version. Please re-read the parts that you have added in the revised manuscript. Sentences are convoluted and it is difficult to follow the ideas.

Some examples:

-Line 81-86 is confusing to the reader, please break the sentences. Too many messages have been packed into two sentences.

- In fig 3, (TCGA data) number of samples need to be pointed out.

-please separate sentence that was added in lines 220-222, its confusing

-Line 229 was significantly associated "with" is missing

-Line 237 References required please, evidences that ADAMs are involved in tumor development and progression

-lines 248-250 is really confusing, please rewrite 

-Lines 250-253, please read the lines carefully, they need language editing.

-Line 330, please replace "as a pilot study, thus study demonstrates" with This pilot study demonstrates" and remove "sufficiently"

Author Response

It seems that the authors have added text in the manuscript in a hurry and sloppiness is obvious when comparing with the first version. Please re-read the parts that you have added in the revised manuscript. Sentences are convoluted and it is difficult to follow the ideas.

Answer) Thank you so much for your insightful and meticulous review. Your advice certainly helped to substantially improve our manuscript. Taking your and other reviewers’ advice very seriously, many parts of the manuscript are rewritten. We, the authors, are very grateful for your time. Amendment of 2nd round review was highlighted in light blue. The revision of the 1st round review stays in yellow. 

Some examples:

-Line 81-86 is confusing to the reader, please break the sentences. Too many messages have been packed into two sentences.

Answer) Thank you so much. We broke the part and reorganized the authors’ idea.

Line 78-87;

Fortunately, sorafenib and regorafenib inhibit the expression of ADAM9 mRNA [24,25] that these drugs may restore the host immunity against HCC and generate a room for synergistic action by adoptive cell therapy with NK cells or CD8+ T cells. Thus, ADAM9-MICA-NKG2D system may provide a strategic target for a novel chemoimmunotherapy combining adoptive NK cell therapy and sorafenib or regorafenib [25].

In this pilot observational study, we aimed to characterize the ADAM9 mRNA expression in blood samples of advanced HCC patients according to their clinical courses. To support our findings, we probed the role of ADAM9 as a prognostic biomarker for HCC using the Cancer Genome Atlas (TCGA) database. Also, we presented a case who achieved complete remission with regorafenib and autologous NK cell combination immunotherapy.

- In fig 3, (TCGA data) number of samples need to be pointed out.

Answer) Thank you so much. Number of samples were noted in the figure and the figure legends.

Line 194-197;

Figure 3. Effect of ADAM9 expression on HCC prognosis in TCGA database. (A) Kaplan–Meier plot of HCC patients (n = 370) according to ADAM9 expression level higher or lower than median (n = 185 for each group). (B) Box-plot comparing ADAM9 expression between HCC primary tumor (n = 370) and adjacent normal liver tissue (n = 50).

-please separate sentence that was added in lines 220-222, its confusing

Answer) Thank you so much. The sentences are separated accordingly.

Line 212-215;

In the present study, we found that the ADAM9 mRNA level in blood was significantly elevated in HCC patients compared with healthy controls. Furthermore, the magnitude of its elevation was much greater in the patients with previous treatment failure than in the newly diagnosed treatment-naïve patients.

-Line 229 was significantly associated "with" is missing

Answer) Thank you so much. It was added.

Line 223;

…higher ADAM9 expression was significantly associated with poor prognosis of HCC.

-Line 237 References required please, evidences that ADAMs are involved in tumor development and progression

Answer) Thank you so much. We added recently published literature regarding ADAMs and HCC. In addition, we summarized those on ADAM9 in Line 256-262 following another reviewer’s advice. The part is included as an answer to your next review request.

Line 243;

… development and progression of HCC [24,25,27-40].

  1. Kohga, K.; Takehara, T.; Tatsumi, T.; Ishida, H.; Miyagi, T.; Hosui, A.; Hayashi, N. Sorafenib inhibits the shedding of major histocompatibility complex class I-related chain A on hepatocellular carcinoma cells by down-regulating a disintegrin and metalloproteinase 9. Hepatology 2010, 51, 1264-1273, doi:10.1002/hep.23456.
  2. Arai, J.; Goto, K.; Stephanou, A.; Tanoue, Y.; Ito, S.; Muroyama, R.; Matsubara, Y.; Nakagawa, R.; Morimoto, S.; Kaise, Y., et al. Predominance of regorafenib over sorafenib: Restoration of membrane-bound MICA in hepatocellular carcinoma cells. J Gastroenterol Hepatol 2018, 33, 1075-1081, doi:10.1111/jgh.14029.
  3. Mazzocca, A.; Giannelli, G.; Antonaci, S. Involvement of ADAMs in tumorigenesis and progression of hepatocellular carcinoma: Is it merely fortuitous or a real pathogenic link? Biochimica et biophysica acta 2010, 1806, 74-81, doi:10.1016/j.bbcan.2010.02.002.
  4. Seals, D.F.; Courtneidge, S.A. The ADAMs family of metalloproteases: multidomain proteins with multiple functions. Genes & development 2003, 17, 7-30, doi:10.1101/gad.1039703.
  5. Xia, C.; Zhang, D.; Li, Y.; Chen, J.; Zhou, H.; Nie, L.; Sun, Y.; Guo, S.; Cao, J.; Zhou, F., et al. Inhibition of hepatocellular carcinoma cell proliferation, migration, and invasion by a disintegrin and metalloproteinase-17 inhibitor TNF484. Journal of research in medical sciences : the official journal of Isfahan University of Medical Sciences 2019, 24, 26, doi:10.4103/jrms.JRMS_129_17.
  6. Shiu, J.S.; Hsieh, M.J.; Chiou, H.L.; Wang, H.L.; Yeh, C.B.; Yang, S.F.; Chou, Y.E. Impact of ADAM10 gene polymorphisms on hepatocellular carcinoma development and clinical characteristics. International journal of medical sciences 2018, 15, 1334-1340, doi:10.7150/ijms.27059.
  7. Li, Y.; Ren, Z.; Wang, Y.; Dang, Y.Z.; Meng, B.X.; Wang, G.D.; Zhang, J.; Wu, J.; Wen, N. ADAM17 promotes cell migration and invasion through the integrin beta1 pathway in hepatocellular carcinoma. Experimental cell research 2018, 370, 373-382, doi:10.1016/j.yexcr.2018.06.039.
  8. Honda, H.; Takamura, M.; Yamagiwa, S.; Genda, T.; Horigome, R.; Kimura, N.; Setsu, T.; Tominaga, K.; Kamimura, H.; Matsuda, Y., et al. Overexpression of a disintegrin and metalloproteinase 21 is associated with motility, metastasis, and poor prognosis in hepatocellular carcinoma. Scientific reports 2017, 7, 15485, doi:10.1038/s41598-017-15800-z.
  9. Liu, Y.; Zhang, W.; Liu, S.; Liu, K.; Ji, B.; Wang, Y. miR-365 targets ADAM10 and suppresses the cell growth and metastasis of hepatocellular carcinoma. Oncology reports 2017, 37, 1857-1864, doi:10.3892/or.2017.5423.
  10. Li, S.Q.; Wang, D.M.; Zhu, S.; Ma, Z.; Li, R.F.; Xu, Z.S.; Han, H.M. The important role of ADAM8 in the progression of hepatocellular carcinoma induced by diethylnitrosamine in mice. Human & experimental toxicology 2015, 34, 1053-1072, doi:10.1177/0960327114567767.
  11. Liu, S.; Zhang, W.; Liu, K.; Ji, B.; Wang, G. Silencing ADAM10 inhibits the in vitro and in vivo growth of hepatocellular carcinoma cancer cells. Molecular medicine reports 2015, 11, 597-602, doi:10.3892/mmr.2014.2652.
  12. Dong, Y.; Wu, Z.; He, M.; Chen, Y.; Chen, Y.; Shen, X.; Zhao, X.; Zhang, L.; Yuan, B.; Zeng, Z. ADAM9 mediates the interleukin-6-induced Epithelial-Mesenchymal transition and metastasis through ROS production in hepatoma cells. Cancer Lett 2018, 421, 1-14, doi:10.1016/j.canlet.2018.02.010.
  13. Hu, D.; Shen, D.; Zhang, M.; Jiang, N.; Sun, F.; Yuan, S.; Wan, K. MiR-488 suppresses cell proliferation and invasion by targeting ADAM9 and lncRNA HULC in hepatocellular carcinoma. American journal of cancer research 2017, 7, 2070-2080.
  14. Wan, D.; Shen, S.; Fu, S.; Preston, B.; Brandon, C.; He, S.; Shen, C.; Wu, J.; Wang, S.; Xie, W., et al. miR-203 suppresses the proliferation and metastasis of hepatocellular carcinoma by targeting oncogene ADAM9 and oncogenic long non-coding RNA HULC. Anti-cancer agents in medicinal chemistry 2016, 16, 414-423, doi:10.2174/1871520615666150716105955.
  15. Zhou, C.; Liu, J.; Li, Y.; Liu, L.; Zhang, X.; Ma, C.Y.; Hua, S.C.; Yang, M.; Yuan, Q. microRNA-1274a, a modulator of sorafenib induced a disintegrin and metalloproteinase 9 (ADAM9) down-regulation in hepatocellular carcinoma. FEBS letters 2011, 585, 1828-1834, doi:10.1016/j.febslet.2011.04.040.
  16. Kohga, K.; Tatsumi, T.; Takehara, T.; Tsunematsu, H.; Shimizu, S.; Yamamoto, M.; Sasakawa, A.; Miyagi, T.; Hayashi, N. Expression of CD133 confers malignant potential by regulating metalloproteinases in human hepatocellular carcinoma. J Hepatol 2010, 52, 872-879, doi:10.1016/j.jhep.2009.12.030.

-lines 248-250 is really confusing, please rewrite 
-Lines 250-253, please read the lines carefully, they need language editing.

Answer) Thank you so much. The part (Line 248-253) was rewritten. We agree that multiple ideas had been mingled. We separated those ideas, added other reviewer’s input, and restructured the corresponding paragraph and the antecedent paragraph (switched the order).   

Line 226-256;

In our study, the change of serum ADAM9 mRNA was easily correlated with clinical response to nivolumab, while the lymphocyte immunophenotype changes were intriguing but not enough to draw a solid conclusion. Namely, Subject #8 did have some favorable changes in her lymphocyte immunophenotypes (mainly PD-1+ or TIM-3+ cytotoxic T cells) but progressed. Another interesting finding was the change of TIM-3 positive helper T cell. Responders (Subject #9 and #10) had it decreased while non-responders (Subject #7 and #8) had it increased. In addition, ADAM9 was strongly correlated with TIM-3 in TCGA database. These findings altogether suggest that there may exist key features in lymphocyte immunophenotype changes that can predict treatment response early on. Our study indicated that the proportion of TIM-3 positive helper T cells may be a good candidate marker, and that unleashing helper T cell from exhaustion may be more important than unleashing cytotoxic T cells. Future studies are needed to elucidate the interplay between the ADAM9-MICA-NKG2D system and lymphocyte immunophenotypes, and to find a relevance between such factors and clinical outcome. 

ADAMs belong to the zinc protease superfamily, and they are usually transmembrane proteins  [27]. Containing disintegrin and metalloprotease domains, ADAMs take part in multiple cellular functions including cell adhesion and migration, proteolysis of the extracellular matrix and shedding of membrane proteins [27,28]. Evidences have indicated that ADAMs are involved in tumor development and progression of HCC [24,25,27-40]. Though the pathogenesis of HCC is multifactorial, it is largely owing to hepatitis B (HBV) or C virus (HCV) infection and alcoholic or non-alcoholic fatty liver. These underlying liver conditions result in chronic inflammation and liver fibrosis causing a continuous remodeling of the extracellular matrix [27]. ADAMs are involved in this inflammatory process that leads to development of HCC [27]. Among several ADAMs associated with HCC, those related with MICA shedding are of note because MICA is a critical part for the cytotoxic cellular immunity. Our study revealed that only ADAM9 had significant association with prognosis of HCC while others, ADAM10 and ADAM17, did not. Regarding ADAM9, previous studies demonstrated that transcriptional suppression of ADAM9 led to inhibition of proliferation and invasion activities of HCC cell lines [24,25,36-40]. Inhibition of ADAM9 protease also showed similar results [41]. Some of these results were backed by the increased mMICA expression and subsequently increased susceptibility of HCC cells to NK cells [24,25,40]. On the other hand, treatment with interleukin (IL)-1β on HCC cell lines increased the expression of ADAM9 and sMICA, and the IL-1β-treated HCC cells became more resistant to the cytolytic activity of NK cells [42].

-Line 326, please replace "as a pilot study, thus study demonstrates" with This pilot study demonstrates" and remove "sufficiently"

Answer) Thank you so much. It was done.

Line 326;

This pilot study demonstrates

Reviewer 3 Report

About the health control in figure 1, have the author also test its expression in man, is there any difference between woman and man?

Could the author double-check the HCC numbers in TCGA dataset for the analysis on page 6?

Page 7, about the correlation between ADAM9 and PD-1, TIM-3, BTLA and LAG-3, the writing might need modification to make the results clear.

Some abbreviation should be kept in uniform, such as quantitative real-time reverse transcription-polymerase chain reaction (quantitative real-time RT-PCR, or real-time q RT-PCR), the author uses quantitative real-time PCR in some other place, though either of them should be fine, it would be better to use them in the same format in one paper.

Did the author also try to explore the role of ADAM9 and immunotherapy in other HCC datasets published by other clinics or basic science research groups?

To the figure 2, could the author add subject # as figure legend on the side of the figure, and how about adding different therapy recycle results in the figure, such as figure 2 c.  to the figure 2c, is there a significant difference between pre-NV and 1st F/U.

To table 2, to the response patient #9, what’s the value used in the table, the 1st F/U or 2st F/U?

Author Response

Answer) Thank you so much for your insightful and meticulous review. Your advice certainly helped to substantially improve our manuscript. Taking your and other reviewers’ advice very seriously, many parts of the manuscript are rewritten. We, the authors, are very grateful for your time. Amendment of 2nd round review was highlighted in light blue. The revision of the 1st round review stays in yellow. 

Q.About the health control in figure 1, have the author also test its expression in man, is there any difference between woman and man?

Answer) Thank you so much. We could not collect men in the control group this time. For controls, we hurried to find banked human blood samples and found samples from women delivered at term. We looked for previous studies if they investigated difference between woman and man. But none of the study reviewed in the following article analyzed difference between woman and man.  

Rinchai D, et al. Increased abundance of ADAM9 transcripts in the blood is associated with tissue damage. Version 2. F1000Res. 2015 Apr 9 [revised 2016 Oct 24];4:89. doi: 10.12688/f1000research.6241.2. eCollection 2015.

Since ADAM9 is affected by medical conditions differently, there must be differences according to age, sex, and underlying medical condition. We think the controls we had for the present study were adequate to provide a starting point to compare with considering the size and aim, a small pilot study to prove concept. We agree with what you meant by the comment. There will be more controls both men and women with controlled age. Thank you so much.

Q.Could the author double-check the HCC numbers in TCGA dataset for the analysis on page 6?

Answer) Thank you so much. There was a mistake. The number of HCC is 370, and that of control (adjacent tissue) was 50. We had mistakenly added those. We corrected the text, figure 3, and the figure legend. Thank you so much for your meticulous review.

Line 186-190; especially the blue high-lights.  

To evaluate effect of ADAM9 expression on HCC prognosis, we performed in-silico analyses with 370 HCC patients from TCGA database. Kaplan-Meier plot revealed that the higher expression group than a median value of ADAM9 expression had significantly poorer overall survival rate (Log-rank test p = 3.9 x 10-4) (Figure 3A). In addition, ADAM9 was significantly upregulated in primary tumor tissues of HCC (n = 370) compared with adjacent normal liver tissues (n = 50) (t-test p = 4.6 x 10-6) (Figure 3B).

Line 194-197;

Figure 3. Effect of ADAM9 expression on HCC prognosis in TCGA database. (A) Kaplan–Meier plot of HCC patients (n = 370) according to ADAM9 expression level higher or lower than median (n = 185 for each group). (B) Box-plot comparing ADAM9 expression between HCC primary tumor (n = 370) and adjacent normal liver tissue (n = 50).

Q.Page 7, about the correlation between ADAM9 and PD-1, TIM-3, BTLA and LAG-3, the writing might need modification to make the results clear.

Answer) Thank you so much. It was rewritten.

Line 200-204;

ADAM9 expression was tested for its correlation with expression of immune checkpoint molecules (PD-1, TIM-3, lymphocyte activation gene-3 (LAG-3) and B and T lymphocyte attenuator (BTLA)) in HCC patients (n = 370) from TCGA database. ADAM9 expression was positively correlated with expression of PD-1, TIM-3 and BTLA, but not with that of LAG-3 (Figure 4). TIM-3 had the strongest positive correlation with ADAM9 (Correlation coefficient r = 0.37 and p = 1.3 x 10-13) (Figure 4B).

Q.Some abbreviation should be kept in uniform, such as quantitative real-time reverse transcription-polymerase chain reaction (quantitative real-time RT-PCR, or real-time q RT-PCR), the author uses quantitative real-time PCR in some other place, though either of them should be fine, it would be better to use them in the same format in one paper.

Answer) Thank you so much. It was corrected accordingly (Line 385; real-time RT-PCR).

Q.Did the author also try to explore the role of ADAM9 and immunotherapy in other HCC datasets published by other clinics or basic science research groups?

Answer) Thank you so much. There are not so many researches done on “immunotherapy using ADAM9” per se. Instead, we added all the researches that we could find, the relevant basic science researches and human researches.

Line 250-256;

Regarding ADAM9, previous studies demonstrated that transcriptional suppression of ADAM9 led to inhibition of proliferation and invasion activities of HCC cell lines [24,25,36-40]. Inhibition of ADAM9 protease also showed similar results [41]. Some of these results were backed by the increased mMICA expression and subsequently increased susceptibility of HCC cells to NK cells [24,25,40]. On the other hand, treatment with interleukin (IL)-1β on HCC cell lines increased the expression of ADAM9 and sMICA, and the IL-1β-treated HCC cells became more resistant to the cytolytic activity of NK cells [42].

Line 273-275;

In CHC patients, serum IL-1β level was positively correlated with serum sMICA level, and serum IL-1β level was significantly higher in CHC patients with HCC than those without HCC [42].

Line 305-307;

In this regard, it is encouraging that one group found an approved drug for anti-alcoholism, disulfiram, and showed that it effectively restored mMICA expression by inhibiting ADAM10 and did not have unfavorable off-target effects [58].

Newly Added References;

  1. Dong, Y.; Wu, Z.; He, M.; Chen, Y.; Chen, Y.; Shen, X.; Zhao, X.; Zhang, L.; Yuan, B.; Zeng, Z. ADAM9 mediates the interleukin-6-induced Epithelial-Mesenchymal transition and metastasis through ROS production in hepatoma cells. Cancer Lett 2018, 421, 1-14, doi:10.1016/j.canlet.2018.02.010.
  2. Hu, D.; Shen, D.; Zhang, M.; Jiang, N.; Sun, F.; Yuan, S.; Wan, K. MiR-488 suppresses cell proliferation and invasion by targeting ADAM9 and lncRNA HULC in hepatocellular carcinoma. American journal of cancer research 2017, 7, 2070-2080.
  3. Wan, D.; Shen, S.; Fu, S.; Preston, B.; Brandon, C.; He, S.; Shen, C.; Wu, J.; Wang, S.; Xie, W., et al. miR-203 suppresses the proliferation and metastasis of hepatocellular carcinoma by targeting oncogene ADAM9 and oncogenic long non-coding RNA HULC. Anti-cancer agents in medicinal chemistry 2016, 16, 414-423, doi:10.2174/1871520615666150716105955.
  4. Zhou, C.; Liu, J.; Li, Y.; Liu, L.; Zhang, X.; Ma, C.Y.; Hua, S.C.; Yang, M.; Yuan, Q. microRNA-1274a, a modulator of sorafenib induced a disintegrin and metalloproteinase 9 (ADAM9) down-regulation in hepatocellular carcinoma. FEBS letters 2011, 585, 1828-1834, doi:10.1016/j.febslet.2011.04.040.
  5. Kohga, K.; Tatsumi, T.; Takehara, T.; Tsunematsu, H.; Shimizu, S.; Yamamoto, M.; Sasakawa, A.; Miyagi, T.; Hayashi, N. Expression of CD133 confers malignant potential by regulating metalloproteinases in human hepatocellular carcinoma. J Hepatol 2010, 52, 872-879, doi:10.1016/j.jhep.2009.12.030.
  6. Itabashi, H.; Maesawa, C.; Oikawa, H.; Kotani, K.; Sakurai, E.; Kato, K.; Komatsu, H.; Nitta, H.; Kawamura, H.; Wakabayashi, G., et al. Angiotensin II and epidermal growth factor receptor cross-talk mediated by a disintegrin and metalloprotease accelerates tumor cell proliferation of hepatocellular carcinoma cell lines. Hepatol Res 2008, 38, 601-613, doi:10.1111/j.1872-034X.2007.00304.x.
  7. Goto, K.; Arai, J.; Stephanou, A.; Kato, N. Novel therapeutic features of disulfiram against hepatocellular carcinoma cells with inhibitory effects on a disintegrin and metalloproteinase 10. Oncotarget 2018, 9, 18821-18831, doi:10.18632/oncotarget.24568.

Q.To the figure 2, could the author add subject # as figure legend on the side of the figure, and how about adding different therapy recycle results in the figure, such as figure 2 c.  to the figure 2c, is there a significant difference between pre-NV and 1st F/U.

Answer) Thank you so much. During this revision, we changed ‘F/U’ to exact number of nivolumab cycles finished before an evaluation. For the Figure 2C, yes, there was a significant difference. We added the 1st follow-up (after 2 cycles of nivolumab) value.

Line 135-140;

Figure 2. Expression level changes of serum ADAM9 mRNA in advanced HCC patients on nivolumab. The number of nivolumab cycles finished before follow-up blood sampling is denoted in the X-axes. After 2 cycles of nivolumab therapy, non-responders did not show significant change in ADAM9 mRNA (A, Subject #7; B, Subject #8). In contrast, responders revealed significant decrease in ADAM9 mRNA (C, Subject #9, from 573.98 + 5.16 to 523.85 + 7.0 (p < 0.05) after 2 cycles, and further down to 262.58 + 20.13 (p < 0.05) after 4 cycles; D, Subject #10, from 323.88 + 10.67 to 85.52 + 5.59 (p < 0.05) after 2 cycles).

Line 151-154;

In Subject #9, the serum ADAM9 mRNA dropped from pre-treatment level of 573.98 + 5.16 to 523.85 + 7.07 (p < 0.05) after 2 cycles, and it further dropped to 262.58 + 20.13 (p < 0.05) after 4 cycles (Figure 2C). In Subject #10, it dropped from 323.88 + 10.67 to 85.52 + 5.59 (p < 0.05, Figure 2D) after 2 cycles of nivolumab.

Q.To table 2, to the response patient #9, what’s the value used in the table, the 1st F/U or 2st F/U?

Answer) Thank you so much. We indicated exact number of nivolumab cycles in Figure 2 and Table 2.

Line 165-166;

Blood samples for the follow-up were acquired after 3 cycles of nivolumab therapy for Subject #7 and 4 cycles for the rest.

Reviewer 4 Report

Comments were taken into account by the Authors of the publication "A Disintegrin And Metalloproteinase 9 (ADAM9) in Advanced Hepatocellular Carcinoma (HCC) and Its Role as A Biomarker during HCC Immunotherapy" . The translation of the lack of adequate funds for scientific research must be left without comment.

Author Response

Comments were taken into account by the Authors of the publication "A Disintegrin And Metalloproteinase 9 (ADAM9) in Advanced Hepatocellular Carcinoma (HCC) and Its Role as A Biomarker during HCC Immunotherapy" . The translation of the lack of adequate funds for scientific research must be left without comment.

Answer) Thank you so much for your insightful and meticulous review. Your advice certainly helped to substantially improve our manuscript. We, the authors, are very grateful for your time. Amendment of 2nd round review was highlighted in light blue. The revision of the 1st round review stays in yellow. Thank you for your time one more time.